# Adapting to Altered Sensory Input: Effects of Induced Paresthesia on Goal-Directed Movement Planning and Execution

**DOI:** 10.3390/brainsci13091341

**Published:** 2023-09-19

**Authors:** Niyousha Mortaza, Steven R. Passmore, Cheryl M. Glazebrook

**Affiliations:** 1Program of Applied Health Sciences, University of Manitoba, Winnipeg, MB R3T 2N2, Canada; 2Faculty of Kinesiology and Recreation Management, University of Manitoba, Winnipeg, MB R3T 2N2, Canada; steven.passmore@umanitoba.ca (S.R.P.); cheryl.glazebrook@umanitoba.ca (C.M.G.)

**Keywords:** induced paresthesia, reaching, goal-directed aiming, somatosensory, kinematic

## Abstract

The current study investigated how temporarily induced paresthesia in the moving limb affects the performance of a goal-directed target aiming task. Three-dimensional displacement data of 14 neurotypical participants were recorded while they pointed to a target on a computer monitor in four conditions: (i) paresthesia-full-vision; (ii) paresthesia-without-target vision; (iii) no-paresthesia-full-vision; (iv) no paresthesia-without-target vision. The four conditions were blocked and counterbalanced such that participants performed the paresthesia and no-paresthesia conditions on two separate days. To assess how aiming performance changed in the presence of paresthesia, we compared early versus late performance (first and last 20% of trials). We found that endpoint accuracy and movement speed were reduced in the presence of paresthesia, but only without target vision. With repetition, participants adjusted their movement performance strategy, such that with induced paresthesia, they used a movement strategy that included more pre-planned movements that depended less on online control.

## 1. Introduction

Sensory input and movement performance are intimately linked for both goal-directed reaching movements and multi-limb coordination tasks [1,2,3,4]. In the natural environment, both visual and proprioceptive feedback are available to plan and update goal-directed reaching movements [5]. The multiple process model of limb control proposes that goal-directed aiming movements are comprised of two components, but multiple processes within these two components [5,6]. According to this model, the first component of the movement is defined by impulse control, which includes the pre-planned part of the movement where the limb covers most of the distance to reach the vicinity of the target. The second component is the limb-target control phase where the available sensory inputs are compared to internal representations. The multiple process model considers multiple factors, such as noise in the neural-motor system, force-related error, as well as efficiency of energy expenditure. It is suggested that the two components of the multiple process model are not independent, and that visual feedback is the main sensory driver of the impulse control phase. Limited visual input leads to decreased accuracy and precision. The availability of visual feedback results in the correction of motor planning errors during the latter phase of the movement trajectory [7].

Proprioception in combination with visual inputs is another major source of information for limb-target regulation. Thus, effective limb control includes integration of expected and actual sensory inputs from multiple sensory sources. There is extensive research on the role of vision in controlling goal-directed movement of the upper limb due to the rich spatial information that vision can provide during movements [8,9,10,11,12,13]. Somatosensory information has a complementary role in the control of human movement by providing information about the location of the limb in the space as well as the relative location of limbs to each other [1,2,3,4,7]. Thus, it is important to know how these two major sources of information are integrated for voluntary limb control. 

Somatosensory input includes sensory input from the body and can be considered as a combination of both tactile and proprioceptive sensory feedback [14,15]. Tactile and proprioceptive inputs are integral for performing everyday tasks, such as grooming, dressing, and meal preparation. Independence may be reduced or lost if the ability to perform functional tasks is altered. For example, if individuals have decreased somatosensory input, they have difficulty reaching for or manipulating an object without having to look at their limb. Indeed, individuals with a variety of sensorimotor disorders that alter the central and/or peripheral nervous system (e.g., stroke, multiple sclerosis, Parkinson’s disease, Diabetes) experience reduced or disrupted somatosensory input. These individuals must then learn or re-learn voluntary goal-directed actions in the presence of altered somatosensory input [16,17]. 

Knowledge about the individual and integrative role of different sensory inputs is crucial for rehabilitation programs when either of these sensory inputs are deficient. Therefore, given the importance of somatosensory input for motor control, and the consequences of its loss, researchers have used a variety of methods to disrupt somatosensory feedback in a neurotypical population to better understand the contributions of somatosensory input for movement control. For example, the natural visual and proprioceptive relationship can be altered by rotating augmented visual feedback [18,19]. Proprioception can be targeted directly using muscle tendon vibration [20,21,22]. While these methods have contributed to our understanding of the role of somatosensory feedback in motor control, the more common sensation of paresthesia experienced by individuals with sensorimotor disorders, and the impact on unconstrained goal-directed movements used in functional tasks, is not well understood [23]. 

Another method to alter somatosensory feedback is to induce paresthesia using transcutaneous nerve stimulation to impair somatosensory inputs [23,24,25]. Paresthesia can be induced by transcutaneous electrical stimulation applied over the estimated path of the chosen peripheral nerve [26,27]. This method does not eliminate the tactile and proprioception inputs, rather it causes a condition that resembles a radiating paresthesia pathology, or a feeling of tingling and numbness radiating along the course of the targeted peripheral nerve. The induced radiating paresthesia resembles the loss of sensory input in a neurologic injury or disease, which makes it a relevant and realistic model.

When performing a variety of motor tasks, including balance and goal-directed reaching tasks, multiple sensory modalities provide input about where and how our limbs are moving. It has been demonstrated that when the preferred modality becomes degraded then the appropriate or preferred modality will shift [8,9,10]. This shift in the preferred sensory modality is consistent with what happens in neurologic injuries or diseases that affect sensory systems. For example, humans become more dependent on vision to maintain balance if the sensitivity of their somatosensory and/or vestibular systems decreases [28,29]. 

However, less is known about how humans compensate for altered or reduced somatosensory input during functional goal-directed movements. Elliott et al.’s multiple process model of goal-directed movement [5] incorporates sensory input from vision and somatosensation. However, the systematic investigation of altered somatosensation, as observed in cases of peripheral nerve damage caused by disease or injury, has been lacking. Given that humans commonly experience changes in afferent inputs, gaining an understanding of the integration between vision, somatosensation, and altered somatosensory inputs is crucial. This knowledge will provide valuable insights into how individuals can adapt their movement performance strategies and the effects of modified somatosensory input.

The specific objectives of the current study were: (1) to determine, in an otherwise healthy nervous system, if the presence of induced paresthesia affects the speed and accuracy of goal-directed movements; (2) to determine if disrupted somatosensory feedback in the effector (index finger and hand) leads to an increased reliance on visual feedback of the target specifically. We hypothesized that individuals will be more reliant on visual feedback in the presence of induced paresthesia, and both the speed and accuracy of goal-directed reaching movements would be affected, and to a greater extent when vision of the target was unavailable [30]. The premise for this manipulation is that when somatosensory feedback is disrupted then, as predicted by the multiple process model, the need for online visual feedback to be available to compare the location of the hand relative to the target location will increase. An additional objective was to explore how participants’ movement strategies may change with experience performing the aiming task in the presence of paresthesia. To carry this out, trials from early and late performance were compared to understand how individuals may adjust their movement control strategies in the context of disrupted somatosensory feedback. Specifically, we hypothesized that participants would update their movement control strategies to include more pre-planned movements that rely less on the need to make online corrections.

## 2. Materials and Methods

### 2.1. Participants

Sixteen neurotypical young adults (ten females, six males), between 18 and 40 years-old, participated in the current study (mean age = 21.7 years, SD = 3.07). All participants were right-handed and reported the absence of any neurological conditions or orthopaedic injuries that might impede their task performance. Furthermore, participants’ visual capacity was established via self-report, where seven participants declared normal vision and ten indicated corrected-to-normal vision. This self-reported information was subsequently confirmed by the experimenter, who also verified the inclusion criteria. Approval by the University of Manitoba Education/Nursing Research Ethics Board was granted, and informed consent was obtained from all participants. This study was carried out in accordance with the ethical standards laid down in the 1964 Declaration of Helsinki.

### 2.2. Task

Participants were asked to aim “as quickly and accurately” as possible to one of four square-shaped targets located on a 17” computer screen that was placed horizontally on a table in front of them (see Figure 1). As described by Fitts’s Law, the index of difficulty (ID) for all targets was six. By selecting an ID of 6, we aimed to provide participants with a task that encouraged them to enhance their performance in terms of both speed and accuracy [31]. An ID of 6 is recognized as a level of difficulty that remains challenging, yet is still attainable. The four possible targets (blue squares) are illustrated in the left panel of Figure 1. In order to attain an ID of 6, we utilized an amplitude of 33 cm in conjunction with a width of 1 cm for the bottom two targets, and for the top two targets, we employed an amplitude of 38 cm along with a width of 1.2 cm. Each trial began with participants holding their index finger on the home position, which was a microswitch located at the bottom of the screen (Figure 1). The experimenter initiated each trial once participants were ready with their finger on the home position. Next, a fixation cross appeared on the screen for a random foreperiod between 800 and1400 ms, after which the target appeared. The target appearing was the signal for participants to move their finger to touch the location of the target on the screen. In the “without target vision” conditions, the target disappeared at movement initiation as measured by the release of the microswitch. Feedback was provided to participants at the end of each trial, including their movement time and if they hit or missed the target. In the “full-vision” condition, participants could see the target throughout the trial. Participants were encouraged to maximize both speed and accuracy through rewards for fast and accurate movements. Specifically, feedback about movement time and target accuracy was provided to encourage participants to challenge themselves to move as quickly and accurately as they could. Participants were rewarded for target hits with movement times of less than 0.5, 1, 1.5, and 2 standard deviations of their baseline movement time (calculation described below), with 4 to 1 draw entries, respectively to win a 100 CAD gift card to the university bookstore [32,33]. 

### 2.3. Experimental Procedure

The critical manipulation for the current experiment was the induction of paresthesia through constant current stimulation of the median nerve [26,34]. In line with similar research, the induced paresthesia method used in the current study followed previous studies. Specifically, the induced paresthesia was applied throughout the testing session that included paresthesia (both with and without target vision). In order to create, or induce, the feeling of paresthesia, a Digitimer DS7AH constant current stimulator (IBIS Instrumentation Canada, Inc., Ottawa, ON, Canada) was used transcutaneously to generate constant stimulation along the median nerve using two disposable and adhesive Ag/AgCl surface electrodes. The median nerve was selected because the areas supplied by this nerve (the index finger and thumb) are critical for fine motor movements. Stimulation electrodes were placed on the frontal aspect of the distal forearm, over the predicted course of the median nerve. Custom E-Prime (version 2.0, Psychology Software Tools, Inc., Sharpsburg, PA, USA) software externally triggered the constant current stimulator, with a pulse duration of 200 μs, an interstimulus interval of 10 ms, and a voltage edge of 0.2 V. To achieve a consistent stimulation level, the level of stimulation was kept at each participant’s premotor threshold. The premotor threshold was established for each participant by systemically establishing the different stimulation threshold levels including sensory, radiating, premotor, and motor. Sensory threshold was defined as the point where participants first reported a change in sensation as a result of the median nerve stimulation. Radiating threshold was defined as the point when participants felt the sensation moving along the median nerve in their forearm. Premotor threshold was the highest stimulation intensity that did not cause any movement, and the motor threshold was the lowest stimulation intensity that caused movement. Disrupted sensation was confirmed immediately prior to performing the test conditions that required paresthesia by assessing the tactile sensitivity of the palmar side of thumb and index fingers using a monofilament light touch test (Touch-Test^®^ Sensory Evaluators: Semmes-Weinstein Monofilaments, North Coast Medical, Inc., Morgan Hill, CA, USA). These results were compared to the results of the baseline monofilament test. 

Experimental procedures are illustrated in Figure 2. Participants attended two testing sessions on two separate days and performed a goal-directed aiming task in one session with induced paresthesia and one session without. Participants started the first session with 20 baseline trials with natural sensory input (full vision of their limb and target and without paresthesia); the baseline data was used to establish the reward-related criteria for the experimental blocks. The experimental conditions included: (i) paresthesia-full vision; (ii) paresthesia-without-target vision; (iii) no-paresthesia-full vision; (iv) no paresthesia-without-target vision. The two experimental sessions were scheduled with a minimum interval of 24 h between them. Specifically, the session involving induced paresthesia (conditions i and ii) was separated by 24 h from the session without induced paresthesia (conditions iii and iv). Additionally, a 5–10 min break was incorporated between the two vision conditions within each daily session. The four experimental conditions and associated target locations were blocked and counterbalanced across participants so that the participants aimed at a unique target location for each condition; however, participants only pointed at one target per block. Testing took place on two separate days; paresthesia was also blocked by day such that on a given day, participants completed both target vision conditions in the specified paresthesia condition. Participants performed 100 aiming trials toward the assigned target for each condition (200 trials per day for a total of 400 trials). Repeated movements to the same target were used to promote participants working to push the limits of how fast they could move while still acquiring the target successfully.

### 2.4. Data Collection and Treatment

Movement characteristics were recorded using a three-dimensional motion capture system (Optotrak 3D Investigator, Northern Digital Inc., Waterloo, ON, Canada). Two infrared emitting diodes (IRED) were taped to the participants’ dominant hand on the dorsal side of the distal phalanx of the index finger (Figure 1, right). Movements were recorded for 2 s for each trial at a sampling frequency of 300 Hz. The motion capture recordings were controlled and synchronized using a custom-made program designed using E-Prime. A customized MATLAB (version: 8.1 (R2013a), the MathWorks, Inc., Natick, MA, USA) program was used to process the raw displacement data acquired by the Optotrak. Movement data were filtered using a 15 Hz dual-pass Butterworth filter. Movement initiation was defined as the first frame that the velocity of the IRED movement went above 30 mm/s and maintained that velocity for 30 ms. Movement completion was detected when the IREDs’ velocity fell below 30 mm/s for 30 ms. The primary axis of movement was defined as the axis with an anteroposterior direction relative to the participant. The secondary axis was established as the mediolateral direction. Both axes were characterized by a positive direction moving away from the body midline. For the primary axis, undershoot errors were defined as movements that were shorter than the target amplitude, and thus closer to the body. For the secondary axis, undershoot errors were similarly defined as movement amplitudes that were shorter than the target location, and thus the actual movement endpoint was closer to the body midline.

### 2.5. Dependent Variables

Temporal dependent measures included reaction time (RT), movement time (MT), and time to peak velocity (ttPV). Reaction time was defined as the time in milliseconds from the “go signal” until movement initiation, while MT was defined as the time in milliseconds from movement initiation until movement termination. In this way, RT provides an index of movement planning and MT provides an index of the time needed to move to the target successfully. Time to peak velocity was measured as the time from movement onset until peak velocity was achieved in the primary axis of movement. Relative time to peak velocity was normalized as a percentage of the MT for that trial. Time to peak velocity and relative time to peak velocity provide an index of the actual and relative time spent in the impulse control phase [35,36]. 

Spatial outcome measures included measurements of movement endpoint accuracy (constant error [CE] and variable error [VE]), and movement trajectories in both the primary (anteroposterior) and secondary (mediolateral) axes. Moreover, VE in the primary axis at 20%, 40%, 60%, 80% of movement time was analyzed as a means to infer online corrections [37,38]. Constant error was used to determine the mean bias participants had about the location of their movement endpoints using the calculation: CE = Σ (x_i_ − T)/n, where x_i_ is the IRED location at the end of the movement on trial i, T is the target location, and n is the number of trials the participant performed. Variable error was used as a measure of within-participant variability in aiming calculated as: VE = (Σ (x_i_ − M)^2^/n)^1/2^, where x_i_ and n are as defined above and M is the mean position of the IRED that the participant reached [39]. In order to investigate the effect of the sensory manipulations (vision and somatosensation) on participants’ ability to improve the performance of their goal-directed aiming movements, only data from the first and last 20 (out of 100 trials) trials in each condition were included in the data analyses and presented as “Early” and “Late” performance. 

### 2.6. Statistical Analysis

In order to establish the efficacy of the induced paresthesia, the outcomes of the monofilament testing were compared before and after induced paresthesia using the Wilcoxon Signed-Rank Test. Dependent variables for movement performance included RT, MT, normalized ttPV, as well as CE and endpoint VE in both primary and secondary axes using a 2 Target Vision (Vision, Without target vision) × 2 Paresthesia (Paresthesia, No-paresthesia) × 2 Time (Early/Late Performance) repeated measures analysis of variance (ANOVA). In order to analyze changes in the spatial variability along the primary (anterior–posterior) axis, a 2 Target Vision (Vision, Without target vision) × 2 Paresthesia (Paresthesia, No-paresthesia) × 2 Time (Early/Late Performance) × 5 Percent of movement completion (20%, 40%, 60%, 80%, 100% of total MT) repeated measures ANOVA was performed. The sample size estimation was performed using G*Power, taking into account an estimated effect size (ηp^2^ = 0.10) derived from previously published studies with similar experimental setups involving reaching movements, both with and without vision (e.g., [40]). With an estimated effect size of 0.10, a desired statistical power level of 0.8, and a significance level set at alpha < 0.05, it was determined that a minimum of 10 participants for the repeated measures ANOVA would be required to achieve the desired statistical power. Significant interactions were further analyzed using Tukey’s HSD post hoc test with alpha = 0.05.

## 3. Results

The Wilcoxon Signed-Rank Test showed that participants required a significantly thicker monofilament to detect skin deformation changes after paresthesia was induced (Z = −3.60, *p* < 0.01). The results of the monofilament testing at baseline and with induced paresthesia are reported in Figure 3. At baseline, all but one of the participants sensed the monofilament size of 2.83, which falls within the normal range based on Touch-Test^®^ thresholds. One participant detected the monofilament size of 3.61, suggesting a decrease in light touch sensitivity. Critical to the experimental manipulation, following induced paresthesia, all participants reported a larger monofilament size, ranging from 3.61 to 4.56 (median = 3.61).

Motion capture data for two participants were excluded from the statistical analysis because displacement data for more than 50% of the trials were missing due to IREDs becoming obscured during the reaching movement. Therefore, data analysis is based on the remaining 14 participants.

### 3.1. Temporal Measurments

For reaction time, no significant main effect or interaction was found for the factors of target vision, time, and paresthesia (results included in Table 1). 

For MT, there were significant main effects for time, F (1, 13) = 15.48, *p* = 0.002, ηp^2^ = 0.54, and paresthesia, F (1, 13) = 8.50, *p* = 0.012, ηp^2^ = 0.39 (Figure 4a). Movement time was shorter without paresthesia than with paresthesia, and during the late 20 trials than the early 20 trials (Table 1). There were no significant interactions between the factors of target vision, time, and paresthesia.

Results of normalized ttPV showed a significant main effect for the factor of Target Vision, F (1, 13) = 11.59, *p* = 0.005, ηp^2^ = 0.47. As shown in Figure 4b, normalized ttPV was longer in without target vision (M = 40.9; SD = 7.1) than full vision conditions (M = 38.5; SD = 6.2). Also, there was a significant interaction between the factors of time and paresthesia, F (1, 13) = 7.25, *p* = 0.018, ηp^2^ = 0.36 (Figure 4c). As illustrated in Figure 4c, Tukey’s HSD test showed that while normalized ttPV for the early and late practice trials was not significantly different, when paresthesia was present, participants had longer ttPV for late trials (M = 40.2; SD = 7.6) versus the early trials (M = 39.3; SD = 5.8) (Figure 4c).

### 3.2. Spatial Measurments

In the primary axis of movement, no significant main effects or interactions were found for the factors of target vision, time, or paresthesia for CE and VE (Table 1). 

In the secondary movement axis, however, there was a significant interaction between the factors of time, paresthesia, and target vision for the outcome of CE, F (1, 13) = 6.03, *p* = 0.029, ηp^2^ = 0.32 (Figure 5a). Post hoc analysis showed that when vision of the target was removed, the CE in early performance was significantly higher when paresthesia was induced (M = −2.2; SD = 2.1) versus without paresthesia (M = −0.2; SD = 2.8).

For VE in the secondary movement axis, there was a significant interaction between the factors of paresthesia and target vision, F (1, 13) = 6.48, *p* = 0.02, ηp^2^ = 0.33 (Figure 5b). Post hoc analysis showed that in the no-paresthesia conditions, participants had significantly higher variability when vision of the target was removed (M = 3.4; SD = 0.9) versus when vision was available (M = 2.9; SD = 0.8). Also, when target vision was not available, participants had significantly higher variability without paresthesia (M = 3.4; SD = 0.9) compared to with paresthesia (M = 3.0; SD = 0.7).

### 3.3. Movement Trajectories

The repeated measures ANOVA for the variability of the movement trajectory in the primary movement axis showed a significant main effect for percent movement completion, F (4, 52) = 87.2, *p* < 0.001, ηp^2^ = 0.87. Also, there was a significant three-way interaction between factors of percent movement completion, time, and paresthesia, F (2.1, 27.3) = 10.74, *p* < 0.001, ηp^2^ = 0.45 (Figure 5c), and another interaction between the factors of time and target vision, F (1, 13) = 13.25, *p* = 0.003, ηp^2^ = 0.50 (Figure 5d). Post hoc analysis for the interaction of time, percent movement completion, and paresthesia showed that during early practice trials when paresthesia was induced, variability was significantly higher compared to the no-paresthesia condition at 20% of the movement; the mean difference of early practice with paresthesia versus no-paresthesia was 5.4 mm. However, this difference was reversed during late trials. Specifically, following more practice without paresthesia, late trials showed significantly higher variability at 20% movement time compared to the paresthesia condition; the mean difference of VE at 20% of the movement for the paresthesia versus no-paresthesia condition for late practice trials was 4.3 mm (Figure 5c). Comparison of the late versus early trials when no paresthesia was induced also showed more variability at late performance (mean difference = 6.5 mm; Figure 5c). Post hoc analysis for the interaction of target vision and time showed that when target vision was available, participants had significantly higher variability in the first 20% of movement time following more practice (late trials) versus early practice trials (Figure 5d). Also, in late practice, the VE at 20% MT was significantly higher with target vision versus without target vision.

Results of the VE for movement trajectory in the secondary axis of movement showed a significant main effect for the factors of time, F (1, 13) = 9.22, *p* = 0.01, ηp^2^ = 0.41 (Figure 5e) and percent movement time, F (2.0, 25.7) = 108.00, *p* < 0.001, ηp^2^ = 0.89 (Figure 5f), indicating higher VE at late versus early performance. Also, there was a significant interaction between percent movement time and target vision, F (4, 52) = 3.01, *p* = 0.026, ηp^2^ = 0.19. Post hoc analysis for this interaction showed that at 60% of movement time, the with target vision conditions had significantly greater variability than without target vision conditions (Figure 5e).

## 4. Discussion

In the current study, we explored if and how participants update their movement control strategies in the presence of induced paresthesia. Participants performed a functional goal-directed aiming task, with or without vision of the target, while paresthesia was induced in order to understand how motor control strategies change when trying to move quickly and accurately while experiencing paresthesia. Neurotypical individuals were recruited to participate in the current experiment and paresthesia was induced using constant current stimulation of the median nerve to assess the contribution of somatosensory input from the hand specifically. 

We hypothesized that both speed and accuracy of goal-directed reaching movements would be affected by paresthesia, and to a greater extent when vision of the target was unavailable for limb-target control mechanisms [8,9,10]. As expected, there was no significant difference between the conditions with and without induced paresthesia when vision of the target was available. However, in the absence of visual feedback, the presence of induced paresthesia had a greater impact on performance compared to the condition without paresthesia. It is well known that limb-target control heavily relies on visual feedback. Therefore, when both somatosensory input was interrupted and vision of the target was removed, the insufficient sensory inputs hindered the ability to make accurate limb-target corrections.

### 4.1. Effect of Sensory Manipulation on Endpoint Accuracy

We predicted that paresthesia would affect movements more robustly when vision of the target was not available. There were no differences in endpoint accuracy in the primary axis of movement. Only the accuracy measures in the secondary movement axis (i.e., mediolateral) were consistent with this prediction. In early performance trials, CE in the secondary axis became significantly larger with paresthesia compared to the no-paresthesia condition, but only when vision of the target was removed. The presence of paresthesia did not have an effect on endpoint accuracy when vison of the target was available. 

Constant error early in performance, without target vision and with paresthesia, was significantly larger than early performance in the paresthesia condition with full vision (Figure 5a). This pattern indicates that visual feedback was the dominant source of feedback for endpoint accuracy as predicted by the multiple process model. Paresthesia without target vision led to participants being consistently biased towards flexion, inferred from the consistent undershoot errors (Figure 5a). One explanation for this bias could be the imbalance in the proprioception caused by stimulation of the median nerve only. Stimulation of the median nerve only could lead to a false sense of location and tendency for overcorrection towards the midline when proprioception was distorted [41]. In a study by Goodman et al. [21], simultaneous muscle tendon vibration of elbow flexors and extensors was used to manipulate somatosensory inputs in neurotypical adults [21]. These authors applied muscle tendon vibration before trials of a horizontal goal-directed aiming task, while vision of the whole environment was manipulated using occlusion goggles. They found that vibration application resulted in increased CE in a goal-directed reaching task when vision was also removed. Their results for CE in the primary axis showed target undershoots, similar to the finding of the current study in the secondary axis. They also found target overshoots in the anterior–posterior axis (secondary axis in that study), similar to findings of the current study for CE in the anterior–posterior axis (which was the primary axis in the current study). It should be noted that the current study was performed in the anteroposterior direction, while the aiming task Goodman et al. used was performed mostly in the mediolateral direction. Although these two studies used different methods for disrupting proprioception, there was a tendency to perceive the position of the upper limb as closer to the midline. Given the location of the stimulations used in these two studies for interrupting proprioception, and the possible tendency to position the upper limb closer to the midline, the consistent findings reinforce the result that interrupting proprioception leads to target undershoots.

Endpoint variability (VE) in the secondary axis was also affected with paresthesia only when vision of the target was removed (Figure 5b). However, in contrast to our expectation, in the no-paresthesia condition specifically, VE was significantly higher when vision of the target was removed compared to when vision of the target was available, namely, induced paresthesia did not increase endpoint variability in the secondary axis. 

### 4.2. Effect of Sensory Manipulation on Movement Strategy

#### 4.2.1. Time to Peak Velocity and Change in Movement Strategy

We found that MT was shorter after performing more trials, which aligns with previous findings [32]. Movement time was also shorter when performing the task without induced paresthesia compared to the induced paresthesia condition. No significant interaction was found for the factors of target vision and paresthesia. Thus, overall MT was impacted by induced paresthesia and improved with practice in all four conditions.

Normalized ttPV was the temporal variable that supported our hypothesis, which predicted that individuals will be more reliant on visual feedback in the presence of a somatosensory disruption. In the current study, when vision of the target was available, regardless of paresthesia condition, ttPV was shorter (see Figure 4b). The changes in ttPV can be interpreted using Elliott et al.’s multiple process model of limb control [5,42]. In this model, there are early and late online control processes: the impulse control and limb-target regulation phases. Most of the impulse control phase happens before the limb reaches peak velocity and is the distance covering a portion of the movement that determines the direction and velocity of the movement. Limb-target regulation happens after the peak velocity is reached and is performed by using the visual and somatosensory inputs to fine tune the landing of the limb on the target. By considering the ttPV finding in the current study, we can interpret that when vision of the target was available, participants reached the proximity of the target quickly and spent more time in limb-target phase, relying on the available visual information of the target location and the limb, whether or not somatosensory input was interrupted [5,40]. However, when vision of the target was removed, the time spent before PV (ttPV) was a larger portion of the movement time. Although statistical analysis did not show a significant effect of paresthesia on the ttPV in the without target vision conditions, analysis of the late versus early performance trials showed that when paresthesia was present, as more trials were performed (comparing first versus last twenty trials), the normalized ttPV increased, namely, a larger percentage of the movement time was spent in the impulse control phase, and less in limb-target regulation. One explanation for this finding could be that participants chose a different strategy for controlling their movement by pre-planning their movements. Pre-planning would be advantageous because they did not have somatosensory input or target vision available for feedback and current control. In other words, with practice, they adopted a strategy that included spending less time on limb-target regulation [32,43]. 

The present findings are consistent with the study conducted by Goodman et al. (mentioned above) [21]. Goodman et al. also found that when vision was removed, or when vibration was applied to interrupt the somatosensory input, participants spent less time after peak velocity for online corrections to the limb movement. 

#### 4.2.2. Spatial Variability and Change in Movement Strategy

According to the findings of the variability of movement trajectories, participants appear to have selected different strategies when performing the task with and without paresthesia. Variability of the movement trajectory early in the movement (~50% of the movement) is known to be associated with motor planning and offline control [38], while spatial variability late in the movement is thought to reflect online corrective processes. Looking at the results of the spatial variability in the primary axis, regardless of the available visual input, when there was no paresthesia, participants were more variable at movement initiation in late trials compared to both early trials without paresthesia and late trials with paresthesia (Figure 5c). Movement initiation was also more variable with paresthesia than without paresthesia for early trials specifically. We propose that when there was no paresthesia, and regardless of the availability of the target, with more practice participants learned to use more open-loop control strategies, including pre-planned initial impulses at movement initiation. This strategy of more forceful initial impulses is expected to lead to higher impulse variability as detected by a larger position variability at 20% of movement time [44]. 

Although participants were asked to move as fast as they could, the average movement time for the task was long enough for participants to perform limb-target corrections and reduce endpoint error and variability (i.e., overall mean of MT = 435 ms is longer than 200 ms, “see Schmidt, 1979 for more information”). On the other hand, when paresthesia was induced, participants appeared to use a more conservative movement strategy after more practice with the task. Specifically, we found lower VE at 20% of movement time during late trials with paresthesia versus early trials (Figure 5c). One explanation for this finding could be that paresthesia may have increased neural-motor noise and affected force specification processes. Thus, participants chose a safe strategy with smaller impulses at movement initiation, leading to less variability at 20% of movement time. The movement time findings are also consistent with this explanation since movements became shorter with practice and longer with paresthesia.

The results of the movement trajectories in the mediolateral axis showed that with more practice, or when vision of the target was available, movements became less variable around 40% and 60% of movement time (Figure 5e,f). This reduced variability of the movement indicates a more pre-planned movement and fewer online corrections when vision of the target was not available.

### 4.3. Effects of Practice 

A secondary objective of the current study was to investigate how ongoing practice with the changed sensory inputs would affect movement strategy and performance. As expected, regardless of experimental condition, participants had shorter MTs with more practice (Figure 4a). Also, normalized ttPV in the paresthesia condition became longer with more practice. These findings indicate that when paresthesia was induced, participants learned to spend more time during the pre-planned impulse control phase, and subsequently fewer online corrections. Although not statistically significant, the trends in movement strategies seemed to be larger when vision of the target was not available (Figure 4b). In summary, the analysis of early versus late trials demonstrated that at least some of the changes in the temporal and spatial movement characteristics that resulted from manipulation of the sensory input were alleviated with practice. 

### 4.4. Target Vision Availability and Movement Strategy 

In the current study, visual input was manipulated by obscuring the vision of the target only because vison of the target is necessary for limb-target control processes that are used to acquire the target accurately [45,46,47]. Lack of target vision (or its memory) is expected to lead performers to use more pre-planned movements as well as using kinesthetic or feedforward sources of information [47]. In the current study, vision of the target was removed upon movement initiation; therefore, the memory of the target location was not decayed and was available for memory guided limb-target regulation [46]. Hence, since target vision was removed in the current study, the effect of paresthesia on execution of a pre-planned and memory guided aiming task was assessed. As expected, the results of this study showed that participants’ aiming accuracy was significantly different in the two without target vision conditions with and without paresthesia (CE in the secondary axis, Figure 5a). A possible mechanism considered for this finding is that the memory-guided movement required more mental effort for the participants and adding induced paresthesia could have overloaded attentional resources, which would interfere with visual attention towards dynamic limb location and online limb-target control processes [48]. It is possible that although vision of the limb was present, and the target location memory should have been available, that participants did not engage online limb-target control processes and used a safe strategy where they pre-planned their movements. The findings of movement trajectories in the secondary axis also support a pre-planned movement strategy when target vision was removed. Removing target vision increased movement spatial variability at 60% of the movement time. Another explanation for the altered movement control strategy when target vision was removed could be that with memory guided movement control participants relied on their perception of the target location [46], which itself was likely distorted or biased as a result of inducing paresthesia by only stimulating the median nerve.

### 4.5. Limitations and Future Directions

One limitation of the current study was the choice of the primary axis of the movement. As observed in the results, median nerve stimulation caused endpoint bias in the mediolateral direction. A task with a mediolateral direction as the primary movement would likely be more sensitive to changes in motor performance as a result of induced paresthesia caused by median nerve stimulation. Similarly, additional insights into the effects of the sensory perturbation on motor control could be assessed with the use of neurophysiological techniques such as transcranial magnetic stimulation (TMS). In the current study, monofilament pressure testing was utilized to assess gross sensory pressure changes as a result of induced paresthesia. However, incorporating a perceptual evaluation of illusory limb location induced by paresthesia, coupled with a measurement of sensory acuity, would have provided a more comprehensive understanding that included both subjective and objective dimensions of the effects of induced paresthesia. This assessment would facilitate a more refined interpretation of the movement accuracy outcomes and a more comprehensive evaluation of the intensity of the induced paresthesia. Future studies will also benefit from extending the current research to encompass neurodiverse populations, such as individuals experiencing typical aging, Down syndrome, or autistic traits. 

## 5. Conclusions

The most consistent finding was that movements performed in the presence of paresthesia took longer to execute but did not take any longer to plan or initiate. Therefore, the reduced movement speed exhibited by participants could be explained by their uncertainty regarding the position of their limb in space. When vision of the target was available, participants’ accuracy and motor control strategies did not change. When vision of the target was not available, paresthesia adversely affected both the accuracy and efficiency of motor performance. We found that participants learned to adapt to the changes caused by induced paresthesia with more practice by pre-planning the movement more, performing fewer online corrections, as well as decreasing their initial movement impulse to compensate for the increased neuromuscular noise. The results of the present study contribute to developing our understanding of how humans modify their motor control strategies when available somatosensory input is disrupted. The present work represents the first step towards extending this line of work to clinical populations who experience disrupted somatosensory feedback. Future work will determine the effect of vision of moving limb and the role of auditory feedback to aid movement performance.

## Figures and Tables

**Figure 1 brainsci-13-01341-f001:**
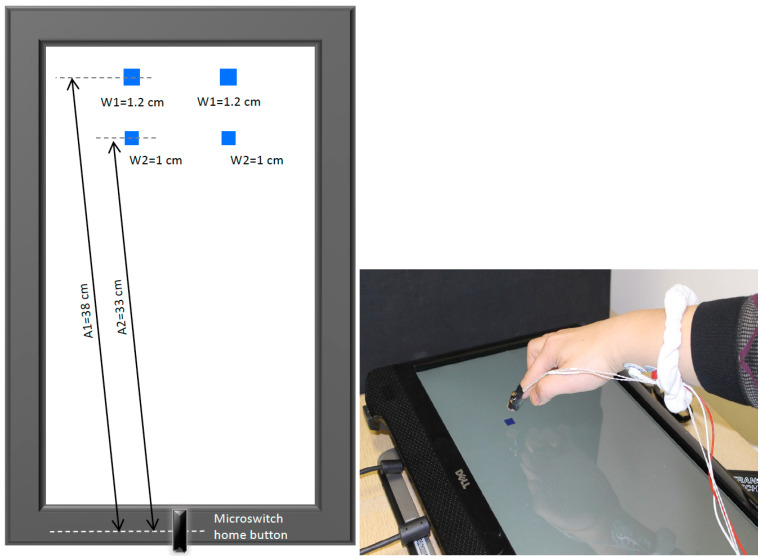
Goal-directed aiming task setup (**right**) and four possible targets (blue squares on **left**) with different amplitudes (A) and widths (W); all targets had an index of difficulty of 6. Note: target amplitude was measured from the centre of the home button to the centre target; one target (i.e., one blue square) was presented per experimental block (see text for details).

**Figure 2 brainsci-13-01341-f002:**
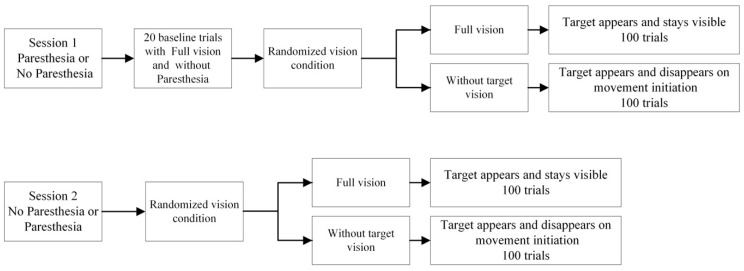
Illustration of the experimental procedure including the two sessions and four experimental conditions.

**Figure 3 brainsci-13-01341-f003:**
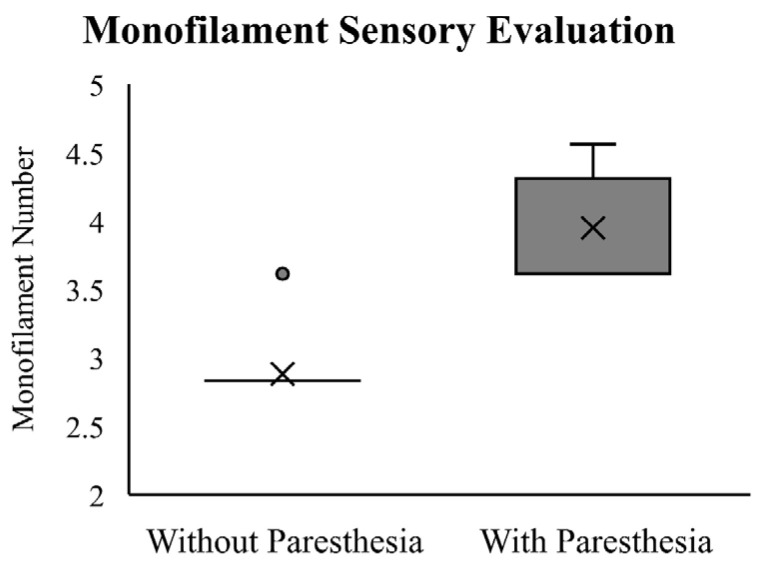
Box plot representing the Touch-Test^®^ Sensory Evaluation—sensed monofilament (grams) before induced paresthesia (Without Paresthesia) and after induced paresthesia (With Paresthesia) as measured prior to the condition with paresthesia. X represents the mean, with interquartile range and maximum values plotted. The dot represents an outlier. Note: Due to a lack of variability at baseline the interquartile range is represented by a line.

**Figure 4 brainsci-13-01341-f004:**
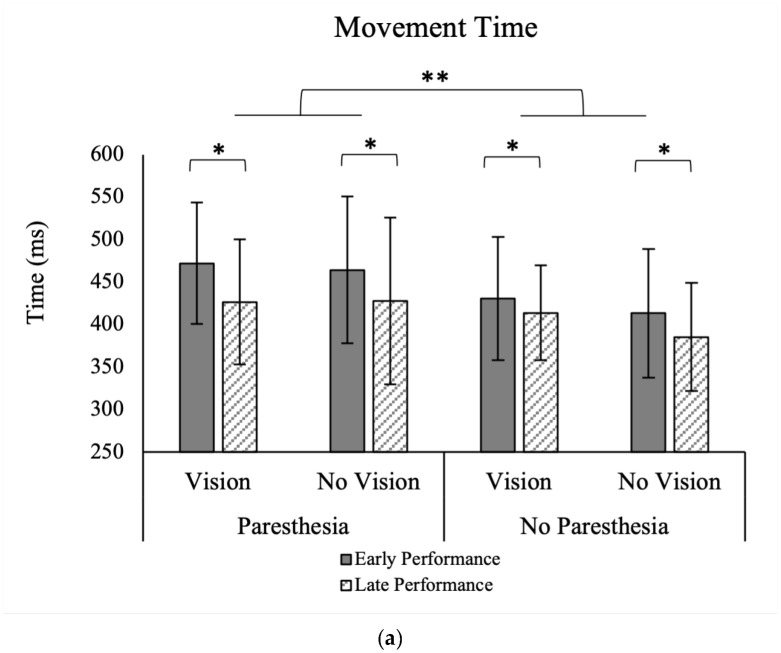
(**a**) Movement time (MT) separated for factors of time, paresthesia, and target vision with significant main effects for the factors of time (*) and paresthesia (**). (**b**,**c**) Normalized percentage of time to peak velocity (ttPV/MT%). (**b**) Time to peak velocity demonstrated separately for different conditions showing a significant main effect for the factor of target vision. (**c**) Interaction of the two factors of time and paresthesia showing significantly higher normalized ttPV with more practice only with paresthesia. All error bars indicate standard error; “No vision” indicates the conditions without target vision.

**Figure 5 brainsci-13-01341-f005:**
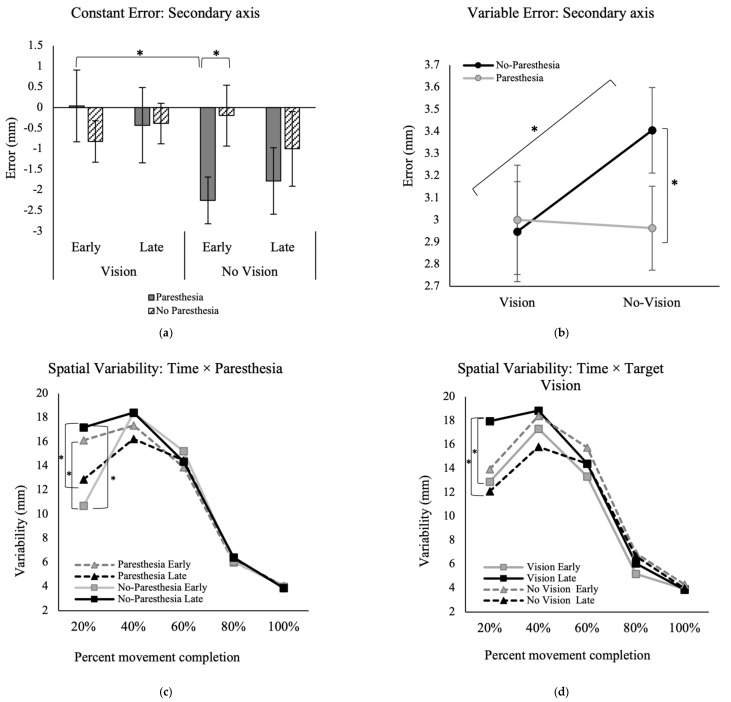
(**a**) The significant interaction of target vision × paresthesia × time for the outcome measure of constant error (CE) in the secondary axis (mediolateral). (**b**) The significant interaction of the two factors of target vision and paresthesia for the outcome of variable error (VE) in the secondary axis. (**c**) The significant interaction of three factors of paresthesia, time, and percent of movement completion. (**d**) The significant interaction of two factors of time and target vision. (**e**) VE in the secondary axis for the significant interaction of target vision and percent movement completion. (**f**) Significant main effect of time for the trajectories in the secondary axis. All error bars indicate standard error; “No vision” indicates the conditions without target vision. * indicates significant differences according to post-hoc analysis (see text for details).

**Table 1 brainsci-13-01341-t001:** Mean and standard deviation (SD) for the temporal outcome measure of reaction time (RT), and raw ttPV, as well as spatial outcomes of variable error (VE) and constant error (CE) in the primary axis of movement for different sensory conditions at early and late trials.

		Paresthesia				No Paresthesia			
		Full Vision		Without Target Vision		Vision		Without Target Vision	
		Early *	Late **	Early	Late	Early	Late	Early	Late
RT (ms)	Mean ± SD	383.0 ± 56.4	375.0 ± 64.6	442.5 ± 169.6	404.7 ± 106.7	379.5 ± 43.5	386.4 ± 80.1	413.5 ± 141.5	377.1 ± 71.8
MT (ms)	Mean ± SD	477.6 ± 70.8	438.5 ± 60.0	464.4 ± 86.5	438.6 ± 92.4	439.8 ± 65.9	414.0 ± 55.8	421.4 ± 71.8	390.6 ± 63.0
ttPV (ms)	Mean ± SD	179.6 ± 35.8	172.0 ± 43.2	180.6 ± 36.0	181.2 ± 34.3	170.4 ± 23.6	157.1 ± 27.6	172.0 ± 33.2	159.1 ± 32.4
VE (mm)	Mean ± SD	3.7 ± 1.0	3.9 ± 0.9	4.3 ± 1.0	4.3 ± 1.4	3.8 ± 1.1	4.0 ± 0.7	4.5 ± 1.1	3.9 ± 1.3
CE (mm)	Mean ± SD	1.3 ± 1.7	1.3 ± 1.2	1.4 ± 2.9	2.0 ± 3.2	1.6 ± 2.6	2.7 ± 1.8	1.8 ± 1.7	2.5 ± 1.7

* Early performance; ** Late performance.

## Data Availability

The data presented in this study are available on request from the corresponding author.

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
