# Peer review of "Adapting to Altered Sensory Input: Effects of Induced Paresthesia on Goal-Directed Movement Planning and Execution"

_brainsci, 2023, doi:10.3390/brainsci13091341_

Round 1

Reviewer 1 Report

The authors provide a sophisticated study design to observe changes in movement control when visual and/or proprioceptive information is absent. Adaptations to these changing constraints were studied comparing the different interventions but also early versus late practice. Not only target achievement but also movement execution over time was studied selecting interesting variables to give insight into changed movement processing. Conclusions are supported by the results revealing adaptation in the interplay of pre-planning and on-line controlling in movement execution to achieve the task goals.

The paper fits well into the scope of the journal. The article is well structured and presents complex results in a comprehensible manner. Before publishing, some corrections should be considered.

main:

Line 163: Is ”premotor threshold” in its first use meant as “pre-session threshold”? Please contrast premotor threshold in lines 162/63 to 167/68, because it seems that the same denotation is used for different things.

At the end of the data collection section please add here also the secondary axes with the definition of plus and minus (overshoot and undershoot) – prepares and makes line 363 easier to understand.

Line 224: the exponent in the formula should be ½ (one half) instead of (-2)

The probability of error cannot be 0.000, therefore please correct to p<0.0005 (exact) or p<0.001 (less decimal places reported) in lines 246, 305, 306, 325. You might add the use of partial eta squared (first noted in line 257) to the statistical analysis section.

Table 1 seems to me too personal and lacks information needed on this personal level. Authors might replace table 1 with a summary table: medians, IQR, min-max of differences – whatever might be interesting or important for special information when paresthesia is induced.

In Figure 3c please correct the title to Time x Paresthesia (leave out “percent” like in d) and add missing legends; in Figure 3f the according title would be Secondary axis Spatial Variability – Target Vision main effect

minor, typos, suggestions:

Line 53: if individuals have decreased …, they have
Line 88: however, less is known
blank in line 226 before [34], – [35] is missing in the running text

3.1. Temporal Measurements (typo)

Authors have introduced abbreviations for reaction time: use RT for line 255 (like MT later on)

Table 2 - Line 271: should this rather be “spatial outcomes”?

Figure 2a: (a) is doubled with different founts – at a first glance confusing

Line 279: if I understood right, this should be factors of time when paresthesia is present?

3.2. Spatial Measurements (typo)

Line 304: maybe you could think of using another abbreviation for the factor “percentage of movement”, earlier in the text addressed as “movement completion percent”, in line 308 as “percent” – could as a proposal be the factor “time completion” (TC)?

References: when citing journal articles sometimes cursive fount but sometimes normal fount appears, please check (e.g. Reference 3, 4 … )

Reviewer 2 Report

I congratulate the authors on the idea. The work is interesting, but I have some comments that need to be resolved.

Major  comments

This work lacks references to the effect of visual input on muscle activity. This information should be added in the introduction. 

L120 – ‘’ (ten females, six males), between 18 and 40 years’’ -The group is quite small, I have a few questions:

·        Has the sample size been counted? If yes provide calculations if no justify why.

·        With such a small group, why didn't they opt for sample heterogeneity?

·        Why was this age range decided upon?

L 121 – ‘’ normal (n=7) or corrected-to-normal (n=10) vision’’

What do the authors mean by this statement? In my opinion, it is an oversimplification. Which needs to be stamped out.

·        What was the visual acuity, what was the best corrected visual acuity? What tool was used to measure it?

·        What was the refractive error ?

·        What was the intraocular pressure?

·        On what basis is it stated that ''corrected-to-normal''-did the authors do a study ?

·        Was there an examination for diseases of the optic nerve / anterior segment of the eyeball ?

L122-123 – ‘’ no neurological condition or orthopaedic injury that would interfere with their task performance.’’ -  Who conducted the orthopedic and neurological examination - what did it look like ?

L179-181 – ‘’ The experimental conditions 1included: i) paresthesia–full vision; ii) paresthesia-without-target vision; iii) no-paresthe- sia-with-full vision; iv) no paresthesia-without-target vision’’ –

·        What was the interval between experimental conditions? 

2.6. Statistical analysis-

·        Add the effect size to the statistically significant results.

Table 1. - Add age and gender in the table.

4. Discussion - Please add information about connections in the neuromuscular pathway that could explain the results obtained

A paragraph of limitations of the study should be added.

In my opinion, the text should also be rewritten to an impersonal form.

Please add a diagram in the methods section to inform about the test procedures. The work is complicated and such a diagram will improve its reading.

Minor comments

L50-58 -Add citations to confirm sentences.

L101 – ‘‘increased reliance’’ -  remove underscore

‘’ Figure 1.’’ - First there should be a reference in the text and then a figure.

L127 – ‘’ 2.2. Task’’  - This comment is not mandatory. The description is correct, however, suggests posting a video of an example of how the experiment was conducted as supplementary material.

In my opinion, this may improve interest in the study.

L156 - Add a photo of what the nduction of paresthesia  looked like.

L162 - -Add citations to confirm sentence.

L246 – ‘’ Z = -3.60, p = 0.00’’ - Standardize the notation and give all results to the second decimal place.

L257 – ‘’ 0.002’’ - Note to all text. - Standardize the notation and give all results to the second decimal place.

L444 – ‘’ [see Schmidt, 1979 for more information]’’ - convert to quote.

Reference analysis

24 papers out of 32 are publications older than 10 years. This accounts for 75% of the work. In my opinion, this is unacceptable. The references should be enriched so that the article is written according to the latest knowledge. Publications older than 10 years can account for a maximum of 40%.

#10 - pages cited are missing.

In references once authors use ''p'' once they do not. This should be standardized. Remove the ''p''.

Self-quotes:

·        Niyousha Mortaza – 0

·        Steven R. Passmore - #18 and # 19

·        Cheryl M. Glazebrook - # 17

Quote 17 occurs three times at work, also individually, in my opinion it is acceptable.  Quote 18 and 19 occur twice in the work and always in the group ''[17-19]'''.  In my opinion, this may be objectionable. In my opinion, should be justify why the authors refer to these works or describe it them more in the paper.

Reviewer 3 Report

This manuscript presents experimental results and analyses when human subjects adapted to disrupted somatosensory inputs during a goal directed aiming task. The main finding is that the presence of induced paresthesia had a greater impact on performance compared to the condition without paresthesia, but only when visual feedback is absent. Such a finding is not starting as the importance of sensory feedback during limb reaching movements has been well recognized, but could still refresh our mind how different modalities of feedback influence the performance of a specific task. Overall, the study was well conducted, the results support the conclusion, and the paper was well organized. Below are some concerns I hope the authors could address.

Please describe the index of difficulty (Line 137, Page 3 and Line 175, Page 4) briefly although you’ve provided a reference. Also, how did the index of difficulty influence the task performing and why did you choose 6?

Figure 1 needs to be improved. The panel on the left looks confusing. Since you plotted a parallelogram to represent the home button, it gives readers an impression that the button was located at a plane perpendicular to the computer monitor, which is not the case according to your descriptions in main texts. Why don’t you use a big rectangle to represent the monitor? You can then plot the targets inside it and the home button outside it and add corresponding annotations, which will be more intuitive for the readers to follow. For the panel on the right, it would be better to pick a different photo showing the arm, hand of the participant and the whole monitor.

Among the four targets two were assigned a width of 1.2 cm, why?

Table 1. There are asterisks above “early” and “late” but I could not find texts explaining them.

Line 226, Page 5, “experiemnta” -> experimental

I don’t think you need to change all “percentage” to “percent”, please check and decide carefully.

Please go through the manuscript to check typos.

Round 2

Reviewer 2 Report

Thank you for sending the revised version of the manuscript. I appreciate the work the authors have put into it. However, I am sorry that I do not accept the current manuscript.

The purpose of the study was ‘’ The specific objectives of the current study were: (1) to determine, in an otherwise healthy nervous system, if the presence of induced paresthesia affects the speed and accuracy of goal-directed movements; (2) to determine if disrupted somatosensory feedback in the effector (index finger and hand) leads to an increased reliance on visual feedback of the target specifically.’’

The goals described above are related to proper visual function, but the authors did not do an expert eye examination.

Earlier, the authors did not check visual acuity. Just because someone wears glasses/lenses does not mean their best corrected visual acuity has reached 20/20. Could be well below 20/20. Visual acuity testing is especially important for vision experiments. Despite correction of the refractive defect, patients may have blurred vision.

In addition, just because a person declares that I do not have a visual defect does not mean that he does not have one.... Unfortunately, a lot of people are not aware of this disease. Does not recognize the answers of the authors:

(Lines 123-128): “All participants were right-handed and reported the absence of any neurological conditions or orthopaedic injuries that might impede their task performance. Furthermore, participants'visual capacity was established via self-report, where seven participants declared normal vision andten indicated corrected-to-normal vision. This self-reported information was subsequently confirmedby the experimenter, who also verified the inclusion criteria.”

In addition, there are ophthalmic diseases that can affect the field of vision. This could have affected the outcome.

In addition, there is no information in the text about the condition of the anterior segment of the eye (such as slit lamp) patients may have had the beginnings of glaucoma/cataracts.

Does not accept answers to the question ‘’Has the sample size been counted? If yes provide calculations if no justify why.’’

‘’ A: We appreciate your input. Power calculations were completed using data from similar experimental designs that involved reaching movements with and without vision. For a statistical power level of 0.8, and alpha <0.05, a minimum of 10 participants was needed to attain appropriate power. Therefore, for the current study we aimed to recruit 12-20 participants in order to insure we have enough statistical power. A sample of sixteen participants was selected specifically because based on the trial types the ideal number of participants were 16 because this would have allowed for a fully balanced design.’’

- ‘’ from similar experimental’’ - What similar studies ? - For future reference, I would ask you to quote.

- What program was used to calculate the power ?

- By what method was this calculated, for any particular test ?

- The calculations should be included in the text.

Yours sincerely
